# Early vs. Late Gestational Diabetes: Comparison between Two Groups Diagnosed by Abnormal Initial Fasting Plasma Glucose or Mid-Pregnancy Oral Glucose Tolerance Test

**DOI:** 10.3390/ijerph192113719

**Published:** 2022-10-22

**Authors:** Tatiana Assuncao Zaccara, Cristiane Freitas Paganoti, Fernanda C. Ferreira Mikami, Rossana P. Vieira Francisco, Rafaela Alkmin Costa

**Affiliations:** 1Departamento de Obstetricia e Ginecologia da Faculdade de Medicina da, Universidade de Sao Paulo, Sao Paulo 05403-000, Brazil; 2Divisao de Clinica Obstetrica do Hospital das Clínicas da Faculdade de Medicina da, Universidade de Sao Paulo, Sao Paulo 05403-000, Brazil

**Keywords:** gestational diabetes, early-onset GDM, oral glucose tolerance test, IADPSG

## Abstract

Gestational diabetes mellitus (GDM) is one of the most common complications in pregnancy. It may be diagnosed using a fasting plasma glucose (FPG) early in pregnancy (eGDM) or a 75-g oral glucose tolerance test (OGTT) (late GDM). This retrospective cohort of women with GDM presents data from 1891 patients (1004 in the eGDM and 887 in the late GDM group). Student’s t-test, chi-squared or Fisher’s exact test and the Bonferroni test for post hoc analysis were used to compare the groups. Women with eGDM had higher pre-pregnancy BMI, more frequent family history of DM, more frequent history of previous GDM, and were more likely to have chronic hypertension. They were more likely to deliver by cesarean section and to present an abnormal puerperal OGTT. Even though they received earlier treatment and required insulin more frequently, there was no difference in neonatal outcomes. Diagnosing and treating GDM is necessary to reduce complications and adverse outcomes, but it is still a challenge. We believe that women with eGDM should be treated and closely monitored, even though this may increase healthcare-related costs.

## 1. Introduction

Gestational diabetes mellitus (GDM) is one of the most common complications in pregnancy, with prevalence varying worldwide according to the criteria used to diagnose this condition [1,2]. It is associated with adverse maternal and perinatal outcomes and comprises different levels of hyperglycemia [2,3]. 

In 2010, the International Association of Diabetes and Pregnancy Study Groups (IADPSG) suggested new criteria to diagnose gestational diabetes [4] based on the Hyperglycemia and Adverse Pregnancy Outcome (HAPO) study [5]. According to their proposal, GDM may be diagnosed using a fasting plasma glucose (FPG)in the first prenatal visit, with no limit to the timing of this initial assessment, or a 75-g oral glucose tolerance test (OGTT) ideally performed between 24 and 28 weeks. 

Their publication also stresses that the decision to perform the initial assessment on all women or only on patients with high risk for diabetes is to be made locally. The OGTT should be offered to all women who did not have GDM diagnosed through an FPG, and not only those with high risk for diabetes, and the proposed thresholds for diagnosing GDM are FPG ≥ 92 mg/dL and/or 1h-OGTT ≥ 180 mg/dL and/or 2h-OGTT ≥ 153 mg/dL. The consensus panel does not recommend performing OGTT before the usual timeframe of 24–28 weeks.

Adopting these criteria increases GDM prevalence and, therefore, increases the cost of healthcare since the follow-up of patients with GDM involves a specialized prenatal team and blood glucose monitoring, among other particularities in managing these cases [4]. Different colleges and associations partially adopted IADPSG proposed criteria. Some use a risk-based approach to offer early testing [2,6], and some suggest different thresholds for considering GDM in the first trimester [6].

Some studies consider FPG levels a risk factor for diagnosing GDM later in pregnancy, but not as a diagnostic method per se [7]. Other studies compare early vs. late GDM pregnancy outcomes [8,9]. However, populations are heterogenous, and results are still conflicting. Most services only consider GDM diagnosis after abnormal OGTT [2,10,11,12]. Therefore, data about early GDM, diagnosed by abnormal FPG, as opposed to OGTT, early in pregnancy, as preconized by IADPSG consensus, are scarce in the literature.

This study aimed to compare clinical and laboratory characteristics and pregnancy outcomes in two subgroups: GDM diagnosed by initial FPG and GDM diagnosed by mid-pregnancy 75 g-OGTT y according to the glycemic level thresholds proposed by IADPSG.

## 2. Materials and Methods

This cohort study was conducted in the Obstetrics Department of Hospital das Clinicas da Faculdade de Medicina da Universidade de Sao Paulo (Sao Paulo—Brazil). We analyzed medical records of all patients with singleton pregnancies seen in our Gestational Diabetes Unit between 1 January 2012and 31 March 2020 and diagnosed with GDM either by initial abnormal FPG done before 24 weeks of gestational age or by abnormal 75 g-OGTT done between 24 and 32 weeks of gestational age, considering glycemic thresholds proposed by IADPSG. We chose the period of 24 to 32 weeks because sometimes, for reasons beyond the control of healthcare providers, women will not test at the recommended gestational age of 24 to 28 weeks. Performing the test at a later gestational age will not minimize the diagnosis, so we decided to use the timeframe used in the HAPO study, which is the study that served as the basis for IADPSG recommendations.

We aimed to evaluate the differences and similarities between both groups of women. This study did not include patients with type 1 or type 2 diabetes diagnoses. Patients with early FPG above 125 mg/dL were not included either.

In the Obstetrics Department, we recommend diagnosing GDM using the flowchart shown in Figure 1.

A multidisciplinary team consisting of physicians, nurses, dietitians and, if needed, social workers, follows all patients diagnosed with GDM in our service. These women receive orientation about lifestyle changes, such as diet and physical activity, and about self-monitoring blood glucose. They are asked to measure blood glucose using a glucose meter at least four times a day. When adequate glycemic levels are not achieved with these interventions only, we use insulin as a first option for pharmacological treatment. Patients with GDM are followed until 39–40 weeks of gestational age. We do not offer elective cesarean section based solely on the diagnosis of GDM.

We recommend an OGTT 6–12 weeks after delivery for all women diagnosed with GDM and use the American Diabetes Association criteria to recognize normal and abnormal results. Impaired fasting glucose is defined as FPG levels of 100–125 mg/dL, and IGT is defined as 2 h-PG levels of 140–199 mg/dL. Diabetes mellitus is diagnosed at FPG ≥ 126 mg/dL or 2 h-PG ≥ 200 mg/dL after a 75-g glucose load.

We reviewed the charts of all patients and collected available information for the following variables: age, prepregnancy body mass index (BMI), parity, history of GDM, family history of DM, connective tissue disease, chronic hypertension, asthma, smoking habit, gestational age at the screening FPG, FPG level at the first appointment, gestational age at the screening OGTT, glucose levels on the OGTT, insulin requirements during pregnancy, gestational age at birth, newborn sex, weight, congenital malformations, stillbirth, 5-min Apgar score and patient’s postpartum OGTT glucose levels.

The study was conducted according to the guidelines of the Declaration of Helsinki and approved by the Ethics Committee of Hospital das Clinicas—FMUSP, Sao Paulo, Brazil (CAAE: 48868915.9.0000.0068). Patient consent was waived due to the retrospective nature of the study

### Statistical Analysis

Patients were classified into two groups according to their diagnostic method: the early GDM group (FPG ≥ 92 mg/dL before 24 weeks of gestational age) and the late GDM group (FPG ≥ 92 mg/dL and/or 1 h-OGTT ≥ 180 mg/dL and/or 2 h-OGTT ≥ 153 mg/dL between 24 and 32 weeks of gestational age). We compared the collected data between the two groups. Quantitative data are shown as means ± standard deviation, or absolute numbers and frequency.

We used Student’s t test, chi-squared or Fisher’s exact test, as appropriate for each analysis. For variables with more than two independent categories, Bonferroni test for post hoc analysis was used. A *p*-value smaller than 0.05 was deemed statistically significant.

Statistical analysis was performed using SPSS version 26 (IBM SPSS Inc., Armonk, NY, USA).

## 3. Results

Overall, 2235 patients were followed-up at the gestational diabetes unit during the abovementioned period; 94 women were carrying twins and therefore were excluded from this analysis. We also excluded 25 patients without a registered date for the diagnostic test and another 225 patients who did the test outside the recommended timeframe (<24 weeks for FPG and between 24 and 32 weeks for OGTT). The remaining 1891 patients were included in the study and were classified into two groups: 1004 in the early GDM group and 887 in the late GDM group.

The baseline and laboratory characteristics of each group are presented in Table 1. Women in the early GDM group had higher pre-pregnancy BMI values (*p* = 0.000), more frequent family history of DM (*p* = 0.003), more frequently reported to have had GDM in a previous pregnancy, and were more likely to have chronic hypertension (*p* = 0.001) and require insulin (*p* = 0.002). They also had higher FPG values and did this test at an earlier gestational age than women in the late GDM group (*p* = 0.000). The late GDM group had more primigravida patients and a more frequent history of connective tissue diseases. There were no statistically significant differences in the frequency of asthma, smoking habit, or fetal malformation diagnosed during prenatal. There were eight cases of miscarriage in the early GDM group.

Birth and newborn data are presented in Table 2. There was a significant difference (*p* = 0.001) in the delivery mode between groups, with patients in the early GDM group being more likely to deliver by cesarean section than women in the late GDM group. There were no statistical differences in stillbirth, gestational age at birth, newborn weight, small-for-gestational-age (SGA), large-for-gestational-age (LGA), or 5-min Apgar below 7. We defined LGA as a birthweight greater than the 90th percentile for the gestational age and small-for-gestational-age (SGA) as a birthweight less than the 10th percentile for the gestational age.

Regarding postpartum evaluation and screening, 1014 patients returned for a follow-up appointment after parturition, corresponding to 53.6% of our cohort. More patients from the early GDM group returned for postpartum screening than from the late GDM group (*p* = 0.023). Three patients had only the 0h-OGTT value, and one had only the 2 h-OGTT value recorded. Of the 1010 women who had a complete puerperal OGTT, the early GDM group was more likely to have impaired fasting glucose or glucose intolerance, while the late GDM group more frequently presented a normal test (*p* = 0.018). There was no statistical difference between DM frequency. There was statistical difference between the mean 0 h-OGTT value for each group (*p* = 0.000) but not between the mean 2 h-OGTT values (*p* = 0.308), as seen in Table 3.

## 4. Discussion

Pregnant women diagnosed with early GDM present different baseline characteristics from those with late GDM, especially concerning features related to metabolic syndrome, such as chronic hypertension and pregestational BMI. Nonetheless, when receiving the same treatment protocol after the early diagnosis of GDM, neonatal outcomes are the same as those found in late GDM. The worse baseline metabolic status can be further verified by a higher prevalence of persistent abnormal glucose metabolism at puerperal OGTT in women diagnosed with early GDM.

There is a diversity of approaches to diagnosing GDM worldwide. Among them, we can highlight the criteria proposed by ADA—which considers GDM the cases of “diabetes diagnosed in the second or third trimester of pregnancy that was not clearly overt diabetes prior to gestation” [13]—and the criteria proposed by IADPSG and endorsed by WHO [14,15] and FIGO [3].

The IADPSG criteria were adopted by several institutions worldwide, including ours. That proposition includes the possibility of diagnosing GDM early in pregnancy through an FPG in the first prenatal visit and later in pregnancy through an OGTT for those patients with a normal early FPG.

Since adopting the IADPSG proposed criteria for diagnosing GDM, healthcare workers identified a new set of patients: the ones with intermediate hyperglycemia early in pregnancy. Their glucose levels are not high enough to be classified as having diabetes in pregnancy, but they reach the threshold for diagnosing GDM later in pregnancy. The clinical significance of identifying early hyperglycemia, and the impact of treating it since early pregnancy, were the object of study for several groups, and the results are conflicting. Extrapolating the glycemic thresholds of OGTT for tests done before 24 weeks of gestational age is controversial, and currently, there is no sufficient evidence to validate its use [16]. The literature also lacks strong evidence on the best way to manage cases of early-pregnancy hyperglycemia [16,17,18]. Moreover, the literature has questioned extrapolating the glycemic thresholds from the OGTT to an isolated FPG during early pregnancy [6,7,9,18,19].

In 2012, Corrado et al. [7] did not see total correspondence between first-trimester FPG ≥ 92 mg/dL and OGTT for diagnosing GDM. Still, they considered the first a highly predictive risk factor for gestational diabetes. Similarly, Zhu et al. [6] in 2013 did not see a complete agreement between early FPG and 24–28 weeks OGTT. They stated that women with FPG ≥ 110 mg/dL should be treated as GDM, while patients with FPG between 92 and 110 mg/dL should not receive a specific diagnosis until the second-trimester OGTT. Nevertheless, they should receive diet and exercise advice from the first trimester. Likewise, Cosson et al. [19] suggested in 2017 an algorithm in which patients with FPG between 92 and 99 mg/dL should not be diagnosed as GDM but should undergo lifestyle changes, such as diet and exercises, and have an OGTT done between 24 and 28 weeks of gestational age. The article describes this intervention as “Prevention of late GDM.” Even though they advise that women should not be diagnosed with GDM, they state that these patients should receive diet and exercise orientation, which is, ultimately, part of the treatment of GDM and may be sufficient for many women to achieve adequate glycemic levels. In the present study, around 80% of the patients had a satisfactory glycemic control with diet, exercises and blood glucose self-monitoring.

Diagnosing and treating milder levels of hyperglycemia early in pregnancy have been the object of analysis by some authors. Cosson et al., in 2021 [18], published an observational study that compared women with different subtypes of hyperglycemia in pregnancy, two of them “early GDM” and “GDM,” according to the same criteria presented in our study. In line with our findings, the authors detected different baseline characteristics between patients with early and late GDM, such as pre-pregnancy BMI and family history of diabetes. Despite both receiving treatment, neonatal outcomes were not different between the groups, in agreement with our results. Other studies demonstrated that women with early-pregnancy hyperglycemia were at increased risk for adverse maternal and fetal outcomes, irrespective of a later second-trimester normal OGTT [20,21,22]. These studies were comprised of women with an abnormal early glucose measurement (each study using a different threshold) that performed an OGTT later in pregnancy. GDM was only diagnosed if the values in the second test were above the defined limits. This is different from what we describe since patients in our service with FPG above 92 mg/dL receive the diagnosis of GDM and do not perform OGTT. Instead, they are readily instructed about diet, exercises and glycemic self-monitoring [17,18,19].

In our population of women with GDM, those with early diagnosis through FPG before 24 weeks had a higher BMI than those diagnosed through mid-pregnancy OGTT. This is comparable with previous findings that increased BMI is a risk factor for early hyperglycemia [7,8,9,19]. The early GDM group presented a more frequent history of diabetes in first-degree relatives and a history of GDM in a previous pregnancy. These characteristics were also described as risk factors for high glucose levels early in pregnancy [7,8,9,19]. Like Bozkurt [8], we found no statistical difference in age between early and late GDM, but in our cohort, women with early GDM reported more frequently having chronic hypertension, similarly to Sweeting [9]. It is essential to highlight that both Bozkurt’s and Sweeting’s studies considered the diagnosis of GDM for women with abnormal results from an early OGTT, not just FPG. Our analysis showed that women in the early GDM group were more likely to need insulin than the ones in the late GDM group. This is similar to Sweeting’s [9] and Cosson’s [18] results and contrary to what Bozkurt described [8]. We found no difference in stillbirth rate, gestational age at birth, birthweight, SGA or LGA rates and 5-min Apgar below 7. The early GDM group had a higher frequency of cesarean section, and the late GDM group had a higher frequency of forceps delivery. This may be due to this group having a higher BMI and a higher frequency of chronic hypertension [23,24,25].

Of note, in our cohort, patients with early GDM followed the same treatment protocol as women with late GDM. Even though they started treatment at an earlier gestational age and were more likely to need insulin, the early GDM group did not present a higher frequency of SGA newborns, suggesting no overtreatment within this group. Similarly, the frequency of LGA newborns was the same between groups. We do not have an untreated early GDM group to compare and verify if this group exposed to untreated hyperglycemia early in pregnancy would present higher frequencies of LGA.

Furthermore, more than 20% of the patients in the early GDM group had an abnormal puerperal evaluation (including IFG, IGT and DM). This group should be targeted as a metabolically susceptible group that should be followed up closely during and after pregnancy. Similarly to Sweeting’s findings [9], more patients in the late GDM group had a normal postpartum OGTT. If women with early GDM had not been identified, closely monitored, and offered postpartum testing, they probably would not be diagnosed with IFG, IGT, or DM and could be susceptible to the long-term effects of untreated hyperglycemia. By recognizing these women early, we could offer healthcare interventions to prevent long-term complications.

Our study’s strength is the large number of cases and the fact that they were treated following the same protocol by the same team throughout the study. We also acknowledge that this study has limitations, such as the relatively small percentage of patients that returned for postpartum testing (around 53%), but this rate is comparable to rates previously described [26,27,28]. Additionally, all our patients are treated for GDM after an early FPG equal to or higher than 92 mg/dL. Therefore, we do not have a comparison group of untreated women to evaluate maternal and perinatal outcomes.

## 5. Conclusions

Diagnosing and treating GDM after an early FPG greater than or equal to 92 mg/dL is controversial in the literature. In our cohort of women diagnosed using this threshold, early treatment resulted in similar neonatal outcomes compared to women diagnosed later in pregnancy. This suggests that there was no overtreatment in the first group.

We recognize that carefully designed prospective studies are still needed to fully understand the impact of detecting early pregnancy hyperglycemia and to determine the best cutoff value for FPG. We hope the present study will help with the discussion on better assisting patients with hyperglycemia during pregnancy.

## Figures and Tables

**Figure 1 ijerph-19-13719-f001:**
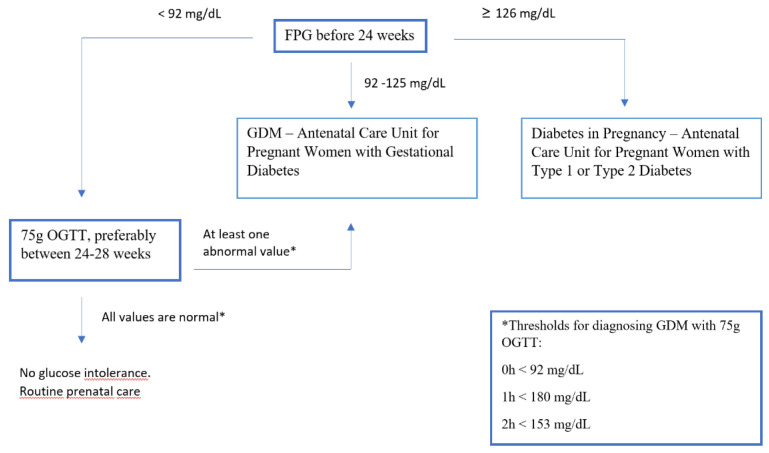
Diagnostic and follow–up flowchart for patients seen in the Obstetrics Department of Hospital das Clinicas da Faculdade de Medicina da Universidade de Sao Paulo.

**Table 1 ijerph-19-13719-t001:** Baseline characteristics of women diagnosed with early or late gestational diabetes.

	Early GDM (FPG before 24 weeks) (n = 1004)	late GDM (75 g OGTT 24–32 weeks) (n = 887)	*p*
Age (years) (n = 1891)	32.6 ± 5.95	32.6 ± 6.21	0.909
BMI (kg/m^2^) (n = 1861)	30.68 ± 6.9	28.08 ± 5.92	0.000 *
Family history of DM (n = 1887)	598/100 (59.4%)	471/887 (53.1%)	0.003 *
Previous GDM status (n = 1883) ^a^			0.000 *
Primigravida	255/998 (25.6%)	266/885 (30.0%)	
Yes	123/998 (12.3%)	63/885 (7.1%)	
No	620/998 (62.1%)	556/885 (62.8%)	
Chronic hypertension (n= 1891)	313/1004 (31.2%)	217/887 (24.5%)	0.001 *
Asthma (n = 1735)	36/1004 (3.6%)	36/731 (4.9%)	0.167
Connective tissue disease (n =1873)	16/1004 (1.6%)	38/869 (4.4%)	0.000 *
Smoking habit (n =1803)	64/921 (6.9%)	50/882 (5.7%)	0.264
Fetal malformation (n = 1638)	50/917 (5.5%)	37/721 (5.1%)	0.774
Gestational age—FPG (n= 1765)	10w2d ± 30 d	12w4d ± 38 d	0.000 *
FPG (mg/dL) (n = 1778)	97.85 ± 6.16	81.03 ± 7.05	0.000 *
Gestational Age—OGTT (weeks + days) (n = 887)	NA	27w0d ± 13d	NA
0 h-OGTT (mg/dL) (n = 887)	NA	90.74 ± 11.50	NA
1 h-OGTT (mg/dL) (n = 866)	NA	164.02 ± 32.77	NA
2 h-OGTT (mg/dl) (n = 869)	NA	153.66 ± 31.44	NA
Insulin use (n = 1811)	211/942 (22.4%)	145/869 (16.7%)	0.002 *
Miscarriage (n = 1602)	8/881	NA	NA

GDM, gestational diabetes mellitus; FPG, fasting plasma glucose; OGTT, oral glucose tolerance test; BMI, body mass index; DM, diabetes mellitus; NA, not applicable. ^a^ Bonferroni Post hoc analysis showed statistical difference between groups in Primigravida and Yes status. * statistically significant

**Table 2 ijerph-19-13719-t002:** Birth and newborn characteristics for women diagnosed with early or late gestational diabetes.

	Early GDM (FPG before 24 Weeks) (n = 1004)	Late GDM (75 g OGTT 24–32 Weeks) (n = 887)	*p*
Stillbirth (n = 1608)	14/875 (1.6%)	6/733 (0.8%)	0.159
Gestational age at birth (n = 1570)	38w0d ± 16 d	38w0d ± 14 d	0.911
Delivery mode (n = 1594) ^a^			0.001 *
Vaginal	250/873 (28.6%)	220/721 (30.5%)	
Forceps	39/873 (4.5%)	63/721 (8.7%)	
Cesarean section	584/873 (66.9%)	438/721 (60.7%)	
Newborn sex = female (n = 1586)	408/868 (47.0%)	362/718 (50.4%)	0.176
Newborn weight (grams) (n = 1584)	3075.75 ± 690.35	3015.91 ± 616.59	0.059
Weight classification (n = 1541)			0.133
SGA	102/855 (11.9%)	94/686 (13.7%)	
AGA	687/855 (80.4%)	555/686 (80.9%)	
LGA	66/855 (7.7%)	37/686 (5.4%)	
Apgar 5 min < 7 (n = 1533)	16/817 (2%)	12/716 (1.7%)	0.68

GDM, gestational diabetes mellitus; FPG, fasting plasma glucose; OGTT, oral glucose tolerance test; SGA, small for gestational age; AGA, adequate for gestational age; LGA, large for gestational age; ^a^ Bonferroni post hoc analysis showed statistical difference between groups in forceps and cesarean section. *statistically significant

**Table 3 ijerph-19-13719-t003:** Postpartum evaluation of women diagnosed with early or late gestational diabetes.

	Early GDM (FPG before 24 weeks) (n = 1004)	Late GDM (75 g OGTT 24–32 weeks) (n = 887)	*p*
Returned for post-partum OGTT (n = 1891)	563/1004 (56.1%)	451/887 (50.8%)	0.023 *
Time of post-partum evaluation (days) (n = 917)	51.53 ± 26	48.51 ± 21	0.065
Complete post-partum OGTT result (n = 1010) ^#,a^			0.018 *
Normal	443/562 (78.8%)	382/448 (85.3%)	
Impaired fasting glucose (IFG) or impaired glucose tolerance (IGT)	106/562(18.9%)	62/448 (13.8%)	
DM	13/562 (2.3%)	4/448 (0.9%)	
Post-partum 0h OGTT (mg/dL) (n = 1013) ^#^	90.12 ± 9.73	85.61 ± 8.97	0.000 *
Post-partum 2h OGTT (mg/dL) (n = 1011) ^#^	108.02 ± 32.03	106.00 ± 30.27	0.308

GDM, gestational diabetes mellitus; FPG, fasting plasma glucose; OGTT, oral glucose tolerance test; ^#^ 3 patients only had the 0 h-OGTT value, and one patient only had the 2 h-OGTT value; ^a^ Bonferroni post hoc analysis showed statistical difference between groups in Normal and IFG-IGT status. * statistically significant

## Data Availability

The datasets generated during and/or analyzed during the current study are available from the corresponding author upon reasonable request.

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
