# Peer review of "Early vs. Late Gestational Diabetes: Comparison between Two Groups Diagnosed by Abnormal Initial Fasting Plasma Glucose or Mid-Pregnancy Oral Glucose Tolerance Test"

_ijerph, 2022, doi:10.3390/ijerph192113719_

Round 1

Reviewer 1 Report

In general, this study examined an important clinical question regarding the diagnosis and management of early GDM. The methods and results section are well presented. However, the introduction section, discussion and conclusion section can be improved. Here are the details:

Major comments: 

1. For the introduction section, comparing to the discussion section, the background section is much less detailed with the current state for science in this topic.

1) For instance, please provide a more complete description of diagnosing early GDM using IADPSG criteria, like the time frame, the threshold.

2) More details are needed for the different approached diagnosing early GDM; no diagnosis with OGTT were mentioned in the background section, but it was discussed later in the manuscript.

3) The knowledge gap is not clearly identified either. Line 46 says this topic is scarce in the literature, but you listed quite a few studies in the discussion section. For instance, how is your study different from Cosson et al. 2021 in line 193? any new contribution from your study?

2. For the discussion section, I find it difficult to follow.

1) Too many paragraphs, some can be grouped together. 

2) Line 218, the early GDM group had a higher frequency of cesarean section compared to the late GDM group. This is one of the main findings, however, there is very minimum discussion on this. 

3) Line 200-202, why these studies have conflicting results with your study? More discussion is needed here. 

3. The conclusion section should be more objective based on key findings of the study. The first sentence is just a broad statement not directly related to this study. Line 245-246 is beyond what you can conclude from this study. It is hard to make this kind of suggestion without a comparison group that was not treated with similar glucose level. Also, there maybe more disadvantages of diagnosing more women early in the pregnancy, including maternal psychological adjustment /mental stress results from the early diagnosis.

Minor comments:

1.     Line 41 – don’t need to be a separate paragraph

2.     Line 127 and 127 – state clearly what is the weight cut-off for this two indicators

3.     Line 170 – this sentence is not clear

4.     Line 175 – citation needed here

Reviewer 2 Report

In this paper, the authors “aimed to compare clinical and laboratory characteristics and pregnancy outcomes in two subgroups: GDM diagnosed by early FPG done before 24 weeks of gestational age and GDM diagnosed by 75g‐OGTT done between 24 and 32 weeks of gestational age according to IADPSG criteria”.

Unfortunately, the aim and the methods are based on wrong criteria:

1)    IADPSG recommends only and universal screening at 24-28 weeks of gestation.

2)    Other criteria, adopted in some countries, recommend a selective screening:

-       at 16-18 weeks for women at high GDM risk and

-       at 24-28 weeks for women at medium risk and for high risk women with negative early test.

Thus, the authors can’t refer to “before 24 weeks” or “between 24 and 32 weeks”.

Besides, a similar paper can’t ignore the paramount studies about the fetal growth:

https://doi.org/10.1016/S2213-8587(20)30024-3.

https://doi.org/10.2337/dc16-0160.

https://doi.org/10.1186/s12916-018-1191-7.

https://doi.org/10.1111/dme.13668.

https://doi.org/10.1016/S2213-8587(20)30189-3.

https://doi.org/10.1155/2020/5393952.

Reviewer 3 Report

Gestational diabetes mellitus (GDM) is one of the most common complications observed during the pregnancy. GDM is associated with increased risk of maternal complications, adverse pregnancy and neonatal outcomes (macrosomy, preeclampsia, intrauterine growth restriction, neonatal hypoglycemia, hyperbilirubinemia, hypocalcemia, respiratory distress syndrome, polycythemia).  

This study aims to compare clinical and laboratory characteristics and pregnancy outcomes in two subgroups: GDM diagnosed by early FPG done before 24 weeks of gestational age and GDM diagnosed by 75g‐OGTT performed between 24 and 32 weeks of gestation.

This study is technically very well-performed, fairy well written but conclusions of the work are not very informative and are rather in line with many previous publications.

It should be explained why in this study the GDM was diagnosed by 75g‐OGTT done between 24 and 32 weeks of gestational age according to IADPSG criteria. According to the IADPSG criteria 75 g OGTT should be performed between 24 and 28 weeks of gestation. Also in Figure 1. we can read about the test in 24-28 weeks of gestation. 

Round 2

Reviewer 2 Report

The only misunderstanding derives by the arbitrary criteria adopted by the authors referring to “before 24 weeks of gestational age and GDM diagnosed by 75g‐OGTT done between 24 and 32 weeks of gestational age according to IADPSG criteria”.

Where do IADPSG criteria indicate “before 24 weeks” and “between 24 and 32 weeks”?

Also, gestational diabetes mellitus is a diabetes diagnosed in the second or third trimester of pregnancy that was not clearly overt diabetes prior to gestation (ADA). This is the current definition of gestational diabetes mellitus. Thus, when the authors generically refer to “before 24 weeks”, they are confusing.

Finally, IADPSG criteria are only proposed criteria. They have not been fully adopted (see ADA recommendations for diagnosis of gestational diabetes).
